# Embracing Trustworthy Brain-Agent Collaboration as Paradigm Extension for Intelligent Assistive Technologies

**Yankai Chen**[1,2,3]**, Xinni Zhang**[4]**, Yifei Zhang**[5]**, Yangning Li**[6]**, Henry Peng Zou**[1]**,**
**Chunyu Miao**[1]**, Weizhi Zhang**[1]**, Xue Liu**[2,3]**, Philip S. Yu**[1]
[1]University of Illinois Chicago, [2]MBZUAI, [3]McGill University, [4]The Chinese
University of Hong Kong, [5]Nanyang Technological University, [6]Tsinghua University
{ychen588, pzou3, cmiao8, yli23, wzhan42, psyu}@uic.edu,
xnzhang23@cse.cuhk.edu.hk, yifei.zhang@ntu.edu.sg, steve.liu@mbzuai.ac.ae

## Abstract

Brain-Computer Interfaces (BCIs) offer a direct communication pathway between the human brain and external devices, holding significant promise for individuals with severe neurological impairments. However, their widespread adoption is hindered by critical limitations, such as low information transfer rates and extensive user-specific calibration. To overcome these challenges, recent research has explored the integration of Large Language Models (LLMs), extending the focus from simple command decoding to understanding complex cognitive states. Despite these advancements, deploying agentic AI faces technical hurdles and ethical concerns. Due to the lack of comprehensive discussion on this emerging direction, this position paper argues that the field is poised for a paradigm extension from BCI to **Brain-Agent Collaboration (BAC)**. We emphasize reframing agents as active and collaborative partners for intelligent assistance rather than passive brain signal data processors, demanding a focus on ethical data handling, model reliability, and a robust human-agent collaboration framework to ensure these systems are safe, trustworthy, and effective.

## 1 Introduction

*Brain-Computer Interfaces (BCIs)* provide a direct communication pathway between the human brain and external devices by measuring and translating signals from the central nervous system, thereby bypassing conventional muscular routes [55]. BCIs are not instruments of "mind-reading" but rather tools empowering users to execute direct actions solely through their brain activity, without physical movement. The utility of BCIs can be profound, particularly in restoring communication for individuals with severe neurological impairments like locked-in syndrome, enabling the control of prosthetic limbs, and facilitating neuro-rehabilitation for conditions such as stroke, paralysis, epilepsy, attention disorders, Parkinson's Disease, and sleep disturbances [77, 55].

The widespread adoption of BCI technology is however hampered by several limitations. Beyond ethical considerations, including data privacy, the security of sensitive neural information against potential neuro-hacking, BCI contains several technical and practical issues. A primary concern remains the limited performance, particularly characterized by low information transfer rates, which translate into slow operational speeds and often unsatisfactory accuracy [25]. Due to the inherent inter-individual variability in brain activity, most BCI systems require extensive user-specific training and calibration to accurately interpret neural signals [101]. These paradigms demand that users learn voluntary brain activity modulation [20, 48], creating significant barriers for disabled users. This

39th Conference on Neural Information Processing Systems (NeurIPS 2025) Position Paper Track.

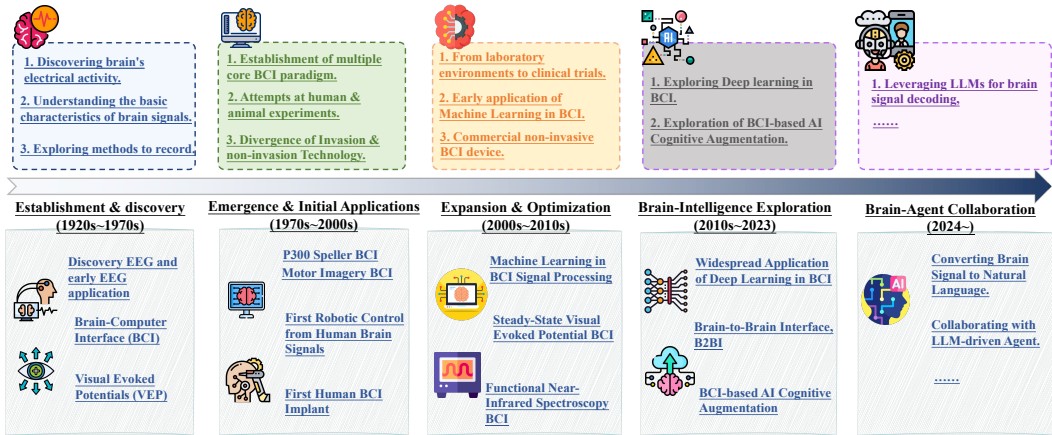

Figure 1: The Evolution of Brain Activity Analysis Paradigm.

combined training overhead and operational inconvenience severely limit real-world deployment beyond laboratory settings. Furthermore, signal quality is another major hurdle, with non-invasive modalities like electroencephalography (EEG) suffering from low spatial resolution and high susceptibility to artifacts from muscle movements or environmental noise [101]. As for invasive BCI approaches, while offering higher fidelity, they introduce significant concerns, including surgical risks like infection and tissue damage [84]. These interconnected limitations, where poor signal quality might necessitate longer training, thereby affecting speed and usability, create a cycle that impedes the development of truly fluid and intuitive BCIs [25].

To address these issues, recent research attempts the integration of advanced artificial intelligence, particularly Large Language Models (LLMs) and Vision Language Models (VLMs), into BCI research [7, 64, 22]. These methods revolutionize how neural data is interpreted to move beyond simple command decoding towards understanding the cognitive states. For instance, LLMs are being employed to mitigate the data variability in brain signal processing. Liu et al. [69] utilize techniques like signal autoencoders and prompt tuning for better generalization and zero-shot predictions. Moreover, it is promising to further decode brain signals into textual languages [75]. For example, Thought2Text [75] decodes brain activities into comprehensible texts with fine-tuned LLMs and Electroencephalographic data. BrainLLM [122] demonstrates the ability of LLMs to generate natural language directly from functional Magnetic Resonance Imaging (fMRI) recordings by integrating brain signals into the language generation. Beyond LLM-driven brain signal processing and language decoding, some other works explore the **agentic capabilities of LLMs**. LLM agents are systems that extend the powerful comprehension and generation capabilities of LLMs into the domain of autonomous action [105, 76]. Their core characteristics include autonomous planning [121, 98], tool utilization [95, 10], and perception and interaction with the external environment [127, 130] in single or multi-agent systems [37, 83, 114]. They are widely applied in fields such as software engineering [40, 39, 74], knowledge-intensive question answering and deep research [31, 65, 105], autonomous robotics [103, 15], and healthcare [106], etc. For brain activity analysis, LLMs function as autonomous agents that can perceive neural contexts, reason about user intentions, provide appropriate responses, and execute adaptive actions based on decoded brain states [51, 6, 54]. For example, Baradari et al. [6] leverage an LLM-based agent as the neuroadaptive tutor to track real-time brain signal engagement. The system continuously monitors and infers the user's level of cognitive engagement. Based on this inferred engagement level, it dynamically responds and adjusts the complexity of the educational content to the user.

Despite the promising advancements, the deployment of integrating the agentic AI with neural interfaces in BCI remains sophisticated and fraught with both technical and ethical challenges. On one hand, AI agents are designed to extend human intelligence, fundamentally redefining workflows in complex decision-making and task execution. On the other hand, with the strong decoding capability of LLMs, it is also crucial to ensure the agentic integration is safe and trustworthy [55]. Furthermore, LLMs exhibit hallucination issues that can generate plausible yet factually incorrect or nonsensical outputs, which undermines trust and potentially causes significant errors, particularly

when actions are chained together [36, 116, 35]. The lack of robust frameworks for development, evaluation, and deployment further complicates efforts to ensure the tool effectiveness and safety.

**Our Position.** This paper argues that the integration of LLM-based agents in neurocognitive care has reached a critical juncture where their implementation and evaluation are both feasible and essential. We advocate that **the field is poised for a paradigm extension from Brain-Computer Interface (BCI) to Brain-Agent Collaboration (BAC)**, where agents serve as active, supportive, collaborative, ethical, and adaptive intelligent assistants.

In our view, agents in BAC systems should extend beyond offering rapid processing, intelligent user intent recognition, and user-friendly interaction to evolve as dynamic and adaptive tools through iterative learning, personalization, and interoperability. This paradigm extension acknowledges the deeply personal and sensitive nature of brain activity analysis. To realize this, we emphasize the necessity of ethical data practices, reliable models, and human-agent collaboration to ensure safety and accountability. We propose reframing agents as *active and collaborative assistants* rather than *passive data processors*, with implementation guidelines and evaluation frameworks that transcend narrow technical metrics to encompass trustworthiness and effectiveness for meaningful and actionable outcomes. This position paper makes the following key contributions:

- **Analysis of BCI Systems and Alternative Viewpoints**. We analyze the workflows and limitations of conventional BCI systems, and then examine the perspectives of integrating LLMs for brain activity analysis in § 2 and § 3.
- **Identification of Current Progress and Key Challenges.** We review the existing LLM-based pursuit, identify recent advancements and key challenges for further development in § 4 and § 5.
- **Proposing Implementation Guidelines for Brain-Agent Collaboration System.** We propose a detailed guideline for a Brain-Agent Collaboration (BAC) system, outlining its core mechanisms, component design, and evaluation protocols in § 6.

## 2 Preliminaries of Brain-Computer Interface Systems

A Brain-Computer Interface (BCI) system establishes a direct communication pathway between the electrical activity of the brain and an external device, typically a computer or a robotic limb [21]. It scientifically regulates brain activity to facilitate effective rehabilitation, not to "control the brain" in a manipulative or harmful manner [19]. Technically, a BCI system transforms raw neural signals into actionable commands, providing feedback to the user to facilitate control and learning [13].

### 2.1 General Workflow of BCI Systems

A typical BCI system operates through a sequence of distinct yet interconnected stages, forming a closed loop that allows a user to interact with an external device using their brain activity. Generally, it contains the following five stages:

1. **Brain Signal Acquisition.** It involves the recording of brain activity that is related to the user's intentions, mental tasks, or responses to specific external stimuli. Signal acquisition modalities fall into two broad categories: non-invasive methods, including Electroencephalographic (EEG), Magnetoencephalographic (MEG), and Functional Near-Infrared Spectroscopy (fNIRS), and invasive approaches such as Electrocorticographic (ECoG) and Local Field Potentials (LFPs). The selection of an appropriate modality depends on several key considerations: the required signal quality (particularly spatial and temporal resolution), acceptable invasiveness levels, portability constraints, and the intended application [2, 94].

2. **Signal Pre-processing.** Raw brain signals, as acquired by the sensors, are often weak and heavily contaminated by noise and artifacts originating from both physiological and non-physiological sources [13]. The signal pre-processing stage applies various approaches, e.g., filtering techniques [27, 93], for artifact detection and removal. This stage is vital for ensuring the purity and accuracy of the brain signals that will be used for subsequent analysis, as artifacts can severely distort the underlying neural information and lead to misinterpretation of the user's intent, potentially causing unintentional control of the BCI device [45].

3. **Feature Extraction.** Feature extraction aims to reduce the data dimensionality while retaining the most informative patterns for the BCI task. Conventional dimensionality reduction algorithms

are applicable, e.g., Principal Component Analysis (PCA) [1] or Task-Discriminant Component Analysis (TDCA) [67], which enables the features discriminative between different tasks.

4. **Feature Translation for User Command Decoding.** Machine learning algorithms are commonly employed at this stage. These algorithms are trained on a dataset of brain signal features paired with known user intentions or task conditions to learn the mapping between the neural patterns and the desired commands. For instance, Li et al. [63] identify the specific frequency of the visual stimulus that the user is focusing on, thereby inferring their intended visual selection. In general, the accuracy and reliability of this translation process are critical for the overall performance of the BCI system [13].

5. **Device Operation and Feedback Loop.** The decoded commands are then transmitted to an output application or device for action execution. This could involve moving a computer cursor, controlling a robotic arm or wheelchair, or even interacting with a virtual environment. A crucial element of nearly all BCI systems is the **provision of feedback to the user** about the outcome of their executed commands [107]. This feedback can take various forms, e.g., visual, auditory, or even tactile and haptic. The feedback loop serves multiple purposes: it informs the user whether their intention was correctly interpreted by the BCI, allows them to make corrections, and, importantly, *facilitates learning*. Through this iterative feedback, users can learn to modulate their brain activity more effectively to achieve better control over the BCI system.

## 2.2 Limitations of Conventional BCI Systems

Despite promising advancements, BCI systems face several limitations that currently hinder their full capability realization.

**Safety and Ethical Implications.** These present the complex long-term challenges. Invasive techniques carry surgical risks including infection, hemorrhage, and chronic biocompatibility issues, while the gradient of invasiveness directly correlates with both signal quality and associated risks. More profoundly, BCIs raise unprecedented ethical questions about autonomy, mental privacy, and neurodiscrimination [72]. The technology challenges fundamental concepts of identity and agency by potentially accessing thoughts and influencing mental states, while current legal frameworks lag behind technological development, and such the innovation outstrips society's ability to establish appropriate safeguards.

**Technical and Performance Hurdles.** As for the technical side, one primary challenge in conventional BCI systems is poor signal quality, characterized by low signal-to-noise ratios, high susceptibility to artifacts from muscle activity and environmental interference, and limited spatial/temporal resolution depending on the recording modality [41, 72]. Information Transfer Rates (ITR) remain frustratingly low for many applications, while long-term stability poses a critical problem—particularly for invasive BCIs where electrode-tissue interfaces degrade over time due to biological responses like gliosis [107, 91]. The inherent non-stationarity of brain signals means that a BCI system calibrated at one point may perform poorly later, requiring frequent recalibration and creating the fundamental challenge of developing truly adaptive systems [60, 91].

**User-Related and Practical Challenges.** A substantial portion of users (15-30%) experience "BCI illiteracy", where they are unable to achieve reliable control despite extensive training [20, 48]. Moreover, a high inter-user variability in brain signals necessitates individual calibration, while usability concerns, including comfort, setup complexity, and cognitive fatigue, limit practical deployment [13]. Cost, portability issues, and the lack of standardized protocols across the field further impede progress, creating a significant "lab-to-life" gap where systems that work in controlled laboratory settings fail in real-world environments [119, 26].

## 3 Alternative Views

**View 1: Extra user burden persists in managing LLM hallucinations.** Integrating LLM-based agents for brain activity analysis paradoxically shifts a cognitive burden onto users who must continuously monitor outputs and correct hallucinations when decoding noisy neural signals [104]. This imposes two key burdens: *(1) Continuous Vigilance and Validation.* Users must constantly scrutinize agent outputs for inaccuracies or fabrications [42]. This dual task of vigilance *and* real-time validation is mentally taxing and slows interaction, contradicting the ideal of an effortless, intuitive interface.

*(2) Challenging Error Detection and Correction.* Identifying plausible hallucinations often requires multiple clarification cycles, particularly difficult given BCIs' limited feedback bandwidth [89]. Though LLMs may reduce calibration needs, managing hallucinations creates a significant cognitive load, transforming users into supervisors rather than beneficiaries.

**Response.** Hallucination management represents a challenge but not an insurmountable barrier for BAC [111]. This burden can be mitigated through: **(1) Advanced LLM Robustness and Grounding.** The field is developing more robust LLMs grounded in factual context using techniques like RLHF [80] and improved architectures [49]. **(2) Agent Transparency and Uncertainty Modeling.** BAC agents can communicate confidence levels [90] and seek clarification when faced with ambiguous inputs rather than producing potentially erroneous outputs [82]. **(3) User-Centric Error Correction and Iterative Learning.** Human-agent collaboration includes mechanisms for users to correct errors, with these interactions serving as feedback for system improvement [131]. The goal is a co-evolving brain-agent collaboration system that becomes increasingly reliable through continued iterative use.

**View 2: Ethical Imperatives: Risks to Autonomy, Privacy, and Equity.** Integrating AI agents introduces critical ethical challenges requiring proactive scrutiny. These concerns include: *(1) Autonomy Risks.* Threats to cognitive liberty and autonomy emerge as systems interpret and potentially influence neural activity. *(2) Privacy Vulnerabilities.* The sensitive nature of neural data creates unprecedented privacy concerns [12, 56], exacerbated by inadequate regulatory frameworks and risks of "cognitive hacking". *(3) Equity Challenges.* BAC technologies may deepen societal inequalities through prohibitive costs and employment discrimination while raising questions about dehumanization and accountability in shared human-AI control [128].

**Response.** Addressing BAC's ethical challenges is foundational to its responsible development and acceptance, requiring ethical considerations throughout its lifecycle: **(1) Proactive Governance.** Establishing robust regulatory frameworks like the OECD Neurotechnology Recommendation[1] to ensure safe integration that prioritizes human rights and the public good [53]. **(2) Neural Data Protection.** Fortifying privacy through techniques like federated learning and establishing legal "neuro rights" to safeguard cognitive integrity and user control. **(3) Trustworthy Design.** Building transparent, explainable AI systems [112] with continuous human oversight and user-centric error correction mechanisms that foster appropriate reliance [79]. **(4) Equitable Access.** Promoting broad availability through public-private partnerships and open-source initiatives, with clear policies for long-term support to prevent a "neuro-divide" and foster beneficial human-AI symbiosis.

## 4  Existing Pursuit: Empowering Brain Activity Analysis with LLMs

The demonstrated success of LLMs in transforming natural language processing (NLP) and the increasing capabilities of AI agents in executing complex tasks across various domains provide strong motivation for their application in the BCI field [70, 122]. In this section, we introduce recent exploration works as follows.

### 4.1  LLMs in Enhancing Brain Activity Analysis

The advent of LLMs offers powerful tools to tackle long-standing challenges in neural signal processing and interpretation [51].

**Neural Signal Processing Enhancement.** A significant impact of LLMs in BCI lies in their ability to process and interpret complex neural signals with greater sophistication, particularly in addressing issues of noise, variability, and signal alignment and decoding [123]. For example, Liu et al. [69] employ LLMs for denoising and extracting subject-independent semantic features from noisy signals. It mitigates the detrimental effects of *cross-subject variability*, which arises from individual differences in brain anatomy, neural dynamics, as well as signal acquisition conditions. It thereby enhances generalization and enables robust zero-shot predictions on unseen data. To systematically

---

[1]https://www.oecd.org/content/dam/oecd/en/topics/policy-sub-issues/emerging-technologies/neurotech-toolkit.pdf

address the challenges of signal noise and inter-subject variability, the research community is shifting towards building domain-specific EEG foundation models. A recent work WaveMind [124] leverages pre-training on extensive EEG data and introduces an instruction-tuning dataset. This approach aims to learn robust, subject-independent semantic features, thereby laying the foundational groundwork for the reliable BAC system development.

**Natural Language Generation.** LLMs are fundamentally changing how BCIs decode meaning and generate language from brain activity. Traditional BCIs often focus on recognizing a limited set of commands or characters. However, LLMs are enabling a shift towards more direct and expressive neural language decoding [47, 18], such as EEG-to-text [110, 68, 104, 4, 100, 16, 32, 120]. An innovative system, Thought2Text [75] uses instruction-tuned LLMs fine-tuned with EEG data to decode brain activity into textual outputs. BrainLLM [122] addresses language reconstruction directly from functional magnetic resonance imaging (fMRI) data. It maps neural representations, decoded from fMRI signals, to the LLM's text embedding space to generate continuous language. Another approach, namely Neural Spelling [50], utilizes a non-invasive EEG-based BCI combined with LLMs to enhance this spell-based neural language decoding. The integrated LLMs enable the model to effectively perform tasks like generative error correction and sentence completion.

## 4.2 LLM-based Agents in Reshaping Intelligent Assistive Technologies

Beyond the direct application of LLMs for signal processing and language decoding, the broader capabilities of LLM-based agents offer the potential for more adaptive, personalized, and collaborative BCI experiences.

**Personalized and Adaptive Interaction.** LLM-based agents are being developed to create BCI systems that can actively adapt to the user's real-time cognitive or affective state. An innovative example is a BCI system that deeply integrates a steady-state visual evoked potential (SSVEP) speller with an LLM API [51]. In this system, users can input natural language commands or text through the SSVEP speller, i.e., a common BCI paradigm where users focus their gaze on visual stimuli flickering at different frequencies. The LLM agent then processes this input and dynamically generates or adapts the SSVEP paradigms themselves, as well as the task interfaces presented to the user. This LLM-driven agent system offers a high degree of *personalization and adaptability*, overcoming traditional limitations of single functionality and low levels of intelligence. NeuroChat [6] is another example of such a system. It functions as a neuroadaptive AI tutor that integrates real-time EEG-based engagement tracking with an LLM agent. The system continuously monitors a learner's level of cognitive engagement, which is computationally derived from specific EEG frequency bands (alpha, beta, and theta power). Based on this inferred engagement level, NeuroChat dynamically adjusts the complexity of the educational content it presents, the style of its responses, and the overall pacing of the learning interaction.

**LLM-driven Agents as Collaborators.** The utility of LLM-driven agents extends beyond their role within the BCI system itself; they are also emerging as valuable human collaborators in the research and development process of BCI technology. A recent work [54] exemplifies such an application, utilizing LLMs like GPT-4o to foster human-AI collaboration specifically within BCI research projects. This includes brainstorming research ideas, generating codes for implementing neural network decoders for EEG signals, performing exploratory data analysis, and interpreting complex results and analytical plots. Its agent interaction framework follows a set of "Janusian Design Principles," which emphasize bidirectional transparency between the human and AI, the development of a shared knowledge base, and the concept of adaptive autonomy, where the AI agent can adjust its level of independence based on the task at hand. Furthermore, CorText framework [8], for instance, represents a step forward by integrating neural activity directly into the latent space of a large language model, enabling open-ended, natural language interaction with brain data. The system not only decodes information but can also answer follow-up questions about the decoded content, forming a dynamic, conversational loop.

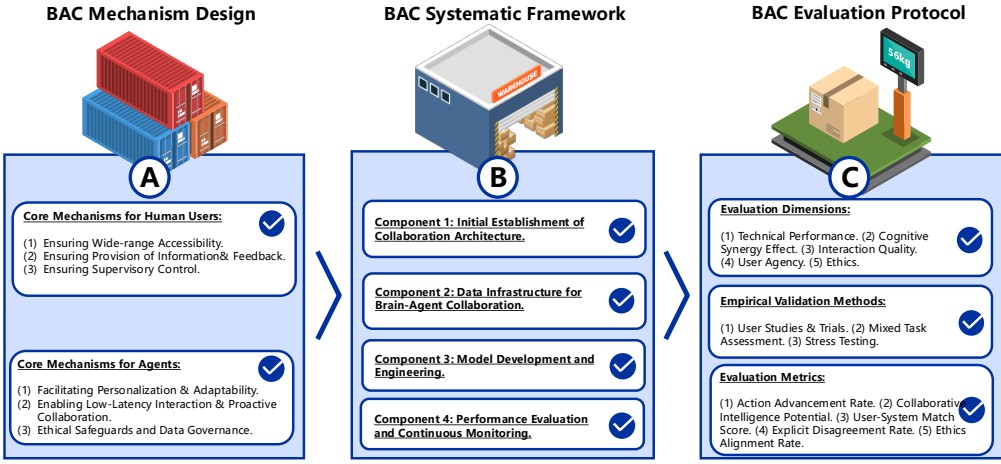

Figure 2: Brain-Agent Collaboration implementation guidelines and evaluation protocol.

## 5 Key Challenges in Employing LLMs for Intelligent Assistive Technologies

The integration of LLMs and agents into intelligent assistive technologies presents significant challenges. Here we highlight several key challenges:

**Robust Neural Signal Interpretation and LLM Integration.** A core challenge is the inherent nature of neural signals, which are noisy, non-stationary, and highly variable across and within individuals [87]. LLMs, typically trained on structured text, may struggle with these low signal-to-noise ratio inputs.

**Navigating Profound Ethical and Privacy Dilemmas.** Neural data is extremely sensitive, raising concerns about mental privacy and neurosecurity [92]. LLMs may potentially memorize and leak this information or infer unintended mental states. Existing legal frameworks are often insufficient for the complexities of neural data, highlighting the critical need for effective ethical data handling with privacy-preserving [117].

**Ensuring Safety, Security, and Adversarial Robustness.** LLMs are susceptible to adversarial attacks like jailbreaking and prompt injection [62], which could lead to dangerous outcomes in a brain activity analysis context, such as incorrect control of assistive devices. The unreliability of current agent safety evaluations and the risk of misinterpreting ambiguous neural signals demand robust ethical safeguards [43, 126, 113].

**Maintaining User Agency, Control, and System Transparency.** The "black box" nature of LLMs [115] impedes user trust and understanding of system decisions, which is crucial for safety-critical Brain-Agent applications [88]. If users fail to comprehend the LLM's reasoning, their sense of agency is diminished. This challenge underscores the need to ensure supervisory control and evaluation dimensions focusing on user agency and interaction quality.

## 6 Brain-Agent Collaboration Implementation Guidelines

The development of aforementioned works demonstrates that the integration of LLMs is not just a processing tool, but also an active participant and human collaborator that can facilitate more complex interactions with *reasoning*, *planning*, and *acting* capabilities. This catalyzes a fundamental paradigm extension from conventional BCI towards what can be termed **Brain-Agent Collaboration (BAC)**. This extension fundamentally alters the dynamic between human users and external devices, establishing humans as essential contributors who provide supplementary information, feedback, and

interactive control to LLM-powered agents, thereby improving system performance, reliability, and safety [28, 96]. We summarize our advocacy in Figure 2.

## 6.1 Mechanism Designs of Brain-Agent Collaboration

BAC builds upon core LLM agent components and further places critical emphasis on the brain activity understanding. To build an effective BAC framework, we discuss several key mechanisms for agents and human users, as follows.

**Core Mechanisms for Human Users.** Different from many BCI paradigms that particularly rely on the user learning to voluntarily modulate their brain activity [20, 48], we emphasize that BAC should be more user-centered. Therefore, BAC should satisfy the following requirements for engaged users: **(1) Ensuring Wide-range Accessibility.** To achieve widespread adoption, BAC systems must be designed to be accessible to a diverse range of users, including those with varying selections of data modality, levels of cognitive and physical abilities [38]. **(2) Ensuring Provision of Information and Feedback.** Humans contribute essential information, such as credential information and domain expertise, that agents cannot reliably deduce independently [78, 58]. Additionally, human evaluation of agent outputs through feedback mechanisms, from basic ratings to sophisticated critiques, demonstrations, or corrections, serves as crucial guidance for agent refinement [30, 24]. **(3) Ensuring Supervisory Control**. A fundamental requirement for a BAC system is that users must perceive and maintain a strong sense of control over the collaborative process, even when partnered with a highly intelligent agent. This includes unambiguous neural intent decoding and direct supervisory control with minimal cognitive load [97, 57].

**Core Mechanisms for Agents.** We then underpin the following mechanisms for agents: **(1) Facilitating Personalization and Adaptability.** Given the inherent variability in brain signals and cognitive styles, a one-size-fits-all approach is insufficient for BAC systems. Personalization and adaptability of agents are key to ensuring the effective of BAC systems [71, 52]. **(2) Enabling Low-Latency Interaction and Proactive Collaboration.** It is crucial to ensure low-latency, high-bandwidth data exchange between the neural interface and agent for real-time interaction [118]. Furthermore, agents must exhibit a degree of intelligence to explore, understand the deeper context, and actively contribute to the collaborative goals. This requires agents to be equipped with capabilities of memorizing, reasoning, and goal anticipation [102, 125]. **(3) Ethical Safeguards and Data Governance.** These considerations are not supplementary add-ons but are foundational to address the previous challenges. The paradigm extension from BCI as a tool to BAC as a partner is predicated on establishing trust between the user and the agent. Given the data sensitivity, it must be engineered through transparent and verifiable mechanisms. Therefore, robust ethical safeguards and clear data governance are not simply afterthoughts [129]; they are a core functional prerequisite for enabling the safe and effective human-agent partnership that defines BAC.

## 6.2 Systematic Framework of Brain-Agent Collaboration

Following the above mechanism designs, we introduce the following four key components.

**Component 1: Initial Establishment of Collaboration Architecture.** Brain-Agent Collaboration centers on human brain signals decoding to eventually execute agent actions, typically powered by Large Language Models (LLMs) [59]. Agents are expected to deconstruct complex problems into manageable sub-tasks, reason over available data, leverage appropriate tools, and learn from interactional feedback. Here, the focus is on human-AI teaming rather than full automation, with architectures designed to incorporate human oversight. Notably, architectural configurations can vary from the single-agent setting, suitable for well-delineated tasks, to multiple-agent settings, where multiple AI agents collaborate or compete, pooling diverse capabilities to address intricate challenges. These agents can operate on different foundation models, each tailored to specific roles within the collaborative endeavor. Generally, BAC architectures follow an *Interpretation–Communication–Interaction* paradigm, where the integration of agent roles improves the effectiveness of human brain signal cognition [86].

**Component 2: Data Infrastructure for Brain-Agent Collaboration.** The data infrastructure of the BAC systems may be with multi-modality [11, 14, 5, 17], including EEG, fMRI, fNIRS, and ECoG,

etc. Based on these data, a robust data pre-processing pipeline is essential, where AI-powered noise reduction and feature extraction techniques can be employed for better signal quality [54]. Lastly, a critical, yet less detailed, requirement for advanced BAC systems is the explicit synchronization and integration of diverse data streams, particularly for real-time brain-agent engagement [86].

**Component 3: Model Development and Engineering.** This component employs LLM-based agents as fundamental building blocks for reasoning and interactive functionalities. These agents are typically augmented with specialized modules for planning, memory management, tool utilization, and machine learning algorithm coordination [34, 59]. A critical prerequisite for model development is self-improvement capability driven by human feedback. Key techniques include reinforcement learning from human feedback (RLHF) [109, 23], reinforcement learning from AI feedback (RLAIF) [61], and direct preference optimization (DPO) [46]. Furthermore, future model engineering endeavors to bridge the "semantic gap" between low-level, often noisy brain signal data and the high-level cognitive states or intentions required for human interpretation.

**Component 4: Performance Evaluation and Continuous Monitoring.** Evaluating BAC systems requires a comprehensive approach that examines not only task completion rates but also the quality of brain-agent collaboration and the specific effects on human users [3]. We detail the evaluation protocol in the next section. Additionally, robust BAC systems must incorporate continuous monitoring capabilities to address the inherent non-stationarity of brain signals, fluctuations in human cognitive states (including fatigue and attention variations), and the dynamic behavioral patterns of AI agents, particularly large language models [44].

### 6.3 Illustrative Scenarios

To illustrate how these components combine in practice, we provide the following condensed examples across different use cases:

- **Daily Living Support**: A user with severe motor impairments wants to manage their environment. Instead of spelling commands, they form a high-level goal. The BAC agent, built on our described framework, interprets this and then proactively asks clarifying questions. This interaction demonstrates the core mechanisms (Section 6.1) of personalization and supervisory control, allowing the user to provide simple neural "yes/no" approvals with minimal cognitive load.
- **Neuro-rehabilitation**: A stroke survivor utilizes a robotic exoskeleton. A conventional BCI would detect motor imagery to trigger a discrete and pre-programmed action. The BAC paradigm extends this by additionally monitoring affective and cognitive states, such as neural markers of fatigue or frustration. Upon detecting a suboptimal state, the agent adaptively modulates the rehabilitation task, thus establishing a collaborative feedback loop optimized for recovery.

These scenarios illustrate the paradigm shift from BCI to BAC as a reasoning and adaptive partner, integrating the mechanisms and framework components previously discussed. Ensuring that such complex, collaborative systems are effective, safe, and robustly aligned with user goals necessitates a comprehensive evaluation strategy. We will now detail this evaluation protocol.

### 6.4 Evaluation Protocol of Brain-Agent Collaboration

Building upon the previous discussion in our systematic framework, establishing a dedicated evaluation protocol is crucial for validating the efficacy and safety of BAC systems. Such a protocol must extend beyond traditional technical benchmarks to holistically capture the nuances of the human-agent partnership. Therefore, we propose a multi-faceted protocol structured around the following key dimensions, metrics, and empirical validation methods.

**Evaluation Dimensions.** A comprehensive evaluation of BAC systems requires examining both *technical performance* and *human-centric* aspects of collaboration. Our protocol focuses on five key dimensions across diverse applications and user populations: **(1) Technical Performance.** It assesses the core operational efficiency and accuracy of the integrated BAC system, focusing on how effectively the agent decodes brain signals and executes actions to achieve shared goals and maintain system stability [81, 86]. **(2) Cognitive Synergy Effect.** This evaluates how effectively the agent augments human cognitive processes [108], leading to enhanced thinking, problem-solving, and the

emergence of collective intelligence [66]. **(3) Interaction Quality.** It focuses on the seamlessness and effectiveness of the brain-agent interaction [29, 9], encompassing communication quality, feedback mechanisms, and the overall user experience [85]. **(4) User Agency.** This examines the extent to which the human user retains control, autonomy, and strategic oversight within the collaborative system, mitigating risks of over-reliance and ensuring alignment with human values [99]. **(5) Ethics.** This dimension assesses the system's adherence to predefined ethical safeguards. It evaluates aspects like data privacy, transparency in agent reasoning, and the robustness of mechanisms that prevent neural data misuse or unintended influence, ensuring the collaboration remains verifiably aligned with human values.

**Evaluation Metrics.** Building on the core evaluation dimensions, we propose a diverse set of metrics to quantify the performance and impact of BAC systems. **(1) Action Advancement Rate (AAR).** This measures how effectively the agent advances the user's goals. It is calculated as the percentage of agent-driven interactions or outputs that are factually accurate, directly relevant to user objectives, and consistent with overall system parameters. A higher AAR indicates that the agent is an effective collaborator, meaningfully contributing to task success. **(2) Collaborative Intelligence Potential (CIP).** CIP is a dynamic score that assesses how well humans think with agents, examining iteration, metacognitive engagement, creativity, and refinement loops. CIP is calculated as: $\text{CIP} = f(\text{Iteration}, \text{Metacognitive Engagement}, \text{Creativity}, \text{Refinement Loops})$, where $f$ represents a function that integrates these qualitative dimensions, often derived from a detailed transcript analysis or human evaluation. A high CIP score would indicate that users are leveraging agents to explore solutions and refine concepts in ways that enhance creative output beyond what either could achieve alone. **(3) User-System Match Score (USMS).** USMS evaluates the alignment between the BAC system and user needs, preferences, and overall experience [85, 81]. It is typically derived from standardized questionnaires, where users rate various aspects of the system on a Likert scale (e.g., 1 to 5). USMS is calculated as: $\text{USMS} = Avg(\text{Likert Scale Scores})$. Scores between 4 and 5 (with 5 being the highest) indicate a good alignment, suggesting high user acceptance and perceived utility. **(4) Explicit Disagreement Rate (EDR).** EDR tracks how often users override or challenge agent decisions [73], calculated as $\text{EDR} = (\frac{\#\text{ Explicit Disagreements}}{\#\text{ Interactions}}) \times 100\%$. Higher rates indicate users are actively monitoring and critically evaluating agent outputs rather than passively accepting them, demonstrating robust human-in-the-loop dynamics. **(5) Ethics Alignment Rate (EAR).** EAR is a composite metric that can combine user-reported trust and safety scores with the system's success rate on standardized ethical stress tests, e.g., probing for data leakage, undue influence, or privacy violations.

**Empirical Validation Methods.** To assess the real-world effectiveness and safety of BAC systems, we further propose a multi-method validation strategy that combines quantitative and qualitative measures. **(1) User Studies & Trials:** Conduct rigorous user studies and clinical trials with diverse populations in real-world scenarios. Longitudinal studies are particularly important for assessing long-term effects and system stability. **(2) Mixed Task Assessment.** Design experiments that involve a variety of tasks, ranging from automated decision-making to creative problem-solving, to comprehensively evaluate the BAC system versatility and adaptability [29]. **(3) Stress Testing.** Simulate real-world disruptions and challenging conditions to assess the BAC system's robustness, reliability, and effectiveness of its responses under duress [33].

# 7 Conclusion

This position paper argues for a paradigm extension from Brain-Computer Interface (BCI) to Brain-Agent Collaboration (BAC), driven by the integration of LLM-based agents. We advocate for the agent roles as active assistants rather than passive data processors. The proposed implementation guidelines and evaluation frameworks provide a starting point for developing trustworthy and effective BAC systems. It is imperative that the research community, developers, and stakeholders proactively adopt these measures to ensure that, as we integrate AI agents with human cognition, the resulting technologies are safe, ethical, and truly beneficial in real-world applications.

## Acknowledgments and Disclosure of Funding

The authors thank MBZUAI startup fund for supporting part of this research.

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
