# OpenReview forum: "Embracing Trustworthy Brain-Agent Collaboration as Paradigm Extension for Intelligent Assistive Technologies"
_NeurIPS.cc/2025/Position_Paper_Track — NeurIPS 2025 Position Paper Track_

### Official Review · Reviewer_JLRQ · 2025-07-22

**Significance:** 3
**Presentation:** 3
**Rating:** 7
**Confidence:** 5

**Summary:**

This paper presents the position that Brain-Computer Interfaces (BCIs), which have recently integrated Large Language Models (LLMs) to address challenges such as low information transfer rates and extensive user-specific calibration, can be extended into a new paradigm: Brain-Agent Collaboration (BAC), driven by LLM-based agents. The authors argue that with LLM integration, agents can be reframed as active and collaborative partners for intelligent assistance, rather than passive brain signal data processors.

The authors identifies challenges faced by conventional BCI systems, including safety and ethical concerns, technical limitations, and user-related challenges. They highlight potential obstacles in implementing the proposed BAC system, such as managing LLM hallucinations a and risks to autonomy and privacy, and proposes solutions to address them. In addition, the authors listed challenges in employing LLMs for intelligent assistive technologies, which includes robust neural signal interpretation and LLM integration among others. Finally, the authors introduce a conceptual framework for BAC and four evaluation metrics to assess its performance.

**Strengths:**

The paper supports its position with clear reasoning and analysis.

**Weaknesses:**

The authors highlights challenges in employing LLMs for intelligent assistive technologies. However they did not provide guidance or suggested solutions for addressing these challenges.

**Questions:**

Have you considered a 5th evaluation dimension for the BAC system focused on Ethics

**Alternative Position:**

Yes, and alternative positions are well-considered and addressed by the argument

**Author Identification:**

No.

**Context:**

3

**Discussion:**

3

**Ethics:**

["NO or VERY MINOR ethics concerns only"]

**Position:**

Yes, the paper argues for or against a position related to machine learning.

**Support:**

3

**Thoroughness:**

5

---

### Official Review · Reviewer_816x · 2025-08-08

**Significance:** 3
**Presentation:** 4
**Rating:** 8
**Confidence:** 5

**Summary:**

The paper proposes Brain Agent Collaboration (BAC), a new paradigm for brain–computer interfaces (BCIs) that embeds large language models (LLMs) into the interaction loop in a proactive, agentic role. It emerges that current BCI systems remain limited, primarily translating brain signals into commands, hence being timely for a shift toward a context-aware interaction. The authors detail shortcomings of existing BCI, including their limited contextual understanding, rigid command mapping, and difficulty in handling signal ambiguity. It then surveys efforts integrating LLMs into BCI that enable models to leverage contextual information and improve interpretation accuracy. Then, the authors address major concerns with LLM–BCI integration, such as privacy risks, susceptibility to hallucinations, and ethical implications of interpreting neural data, outlining mitigation strategies aimed at ensuring safe and responsible deployment. The work ends proposing BAC ecosystem, starting from its core components and extending to evaluation protocols and metrics. The discussion is grounded in an extensive review of the state of the art, positioning BAC as both a technically feasible and socially conscious direction for the future of BCI.

**Strengths:**

+ The paper is well written, clearly structured, and easy to follow.
+ The proposal is timely and highly relevant to the community.
+ The BAC concept is exciting and thoroughly documented.
+ The position is well supported by extensive and pertinent references to current literature.

**Weaknesses:**

- The position is overly optimistic given the significant unresolved issues in both BCI and LLM technologies; current methods are not yet ready for such a shift. Some claims overlook the real limitations of BCI, presenting systems that do not truly work in practice as viable for this transition.
- Section 5 largely repeats concepts from the introduction and could be removed to expand Section 6, allowing a clearer presentation of the full proposal.
- Figures are poorly described and add little value to supporting the position.
- Section 6 lacks concrete examples to effectively illustrate the proposed ecosystem.

**Questions:**

1 - Given the current technological limitations of both BCI and LLMs, how do you envision bridging the gap between the present state of the art and the readiness required for the BAC framework to be feasible in practice?
2 - Could you provide a more critical assessment of the real-world performance of existing BCI systems, particularly addressing those that are currently overrepresented as functional but may not yet deliver reliable outcomes? (e.g., discrepancies in measuring performances with the Leave One Out vs the Leave One Subject Out evaluation strategy)
3 - Can you add concrete, scenario-based examples in Section 6 to illustrate how the proposed BAC ecosystem would operate in practice across different use cases?

**Alternative Position:**

No

**Author Identification:**

No.

**Context:**

4

**Discussion:**

4

**Ethics:**

["NO or VERY MINOR ethics concerns only"]

**Position:**

Yes, the paper argues for or against a position related to machine learning.

**Support:**

3

**Thoroughness:**

5

---

### Official Review · Reviewer_4VHB · 2025-09-02

**Significance:** 4
**Presentation:** 2
**Rating:** 7
**Confidence:** 4

**Summary:**

This paper argues that LLMs present a new suite of mechanisms that could extend the paradigm of Brain-Computer Interfaces (BCIs) from passive data processing to active, collaborative agents, in the form of Brain-Agent Collaboration (BAC). The paper lays out existing research indicating the potential and proximity of this approach and then sets out a framework for realizing BAC systems, including mechanisms for evaluating these systems to ensure accuracy and quality of user experience.

**Strengths:**

This paper effectively communicates foundational information about BCIs, illustrates examples of LLMs in action in the field today, and recognizes potential concerns. They pull citations from a variety of recent publications, demonstrating the timeliness of these questions to the research community. (So fresh that at least one of their cited papers, "LLMs Help Alleviate the Cross-Subject Variability in Brain Signal and Language Alignment," was withdrawn by its authors since submission!) Given the growing interest in agents/agency in AI research, this feels like an appropriate topic for the NeurIPS community to discuss.

**Weaknesses:**

While there are some occasional typos (often missing words or letters), the piece is pretty readable, even to a non-BCI-expert. As a skeptical reader, not all of my concerns were addressed, but some of that may be a matter of the structure or organization of the arguments. The Alternative Views, for instance, do not seem to address the argument head-on, while the evaluation protocol feels a little tacked on (which is a shame because I think it offers some useful examples for the work, but I wish that there was more of a build up to it from the beginning). It feels like it jumps around a bit. The paper would benefit from more signposting (giving the readers a sense of direction) and more reinforcement of the core thesis (why does THIS evidence or section support your ultimate point?)
Early in my reading, I felt like the author(s) handwaved a bit when saying "These [ethical] considerations cannot be afterthoughts in BAC system development. They must be woven in..." In reality, I think that these concerns are at least partially spoken to in the BAC framework and evaluation metrics, but I wish that had been stated more clearly and the flow of the document to make things more obvious.

**Questions:**

Is the main argument you are making that we need this framework (I agree) or that LLMs ought to be used in this collaborative fashion (I'm not sure I'm convinced)?
What entities (people/orgs) do you think need to come to the table to advance the framework you've put forward?
How do you evaluate the tradeoff of risks and benefits in privacy and human agency? Are there any existing frames for this?

**Alternative Position:**

Yes, and alternative positions are well-considered and addressed by the argument

**Author Identification:**

No.

**Context:**

4

**Discussion:**

4

**Ethics:**

["NO or VERY MINOR ethics concerns only"]

**Position:**

Yes, the paper argues for or against a position related to machine learning.

**Support:**

3

**Thoroughness:**

3

---

### Note · Authors · 2025-09-03

**1-10 Additional Comments:**

# Reviewer 816x:
Q1: (1) high-quality signal acquisition, particularly for non-invasive cases; (2) techniques such as multimodal information fusion, continuous tracking and real-time processing capabilities; (3) infrastructure such as recent bidirectional adaptive interfaces will help.
Q2: Most approaches use LOO; LOSO may show performance degradation from within-subject overfitting: lab-to-real-world gaps exist (P300: 90-98% lab vs. 70-80% practical).

# Reviewer 4VHB:
Q1. (1) Our primary position is leveraging LLM-based agents to enhance traditional BCI systems. (2) Individuals contribute via technical innovations; orgs establish governance frameworks/standards. (3) Privacy should be emphasized more as an intelligent assistive technology that genuinely protects/serves users. (4) Current work focuses on task completion; frameworks balancing privacy protection with complex execution remain underdeveloped.
In addition, we'll update the withdrawn paper citation. Thank you.

**1-11 Submit Again:**

Probably yes

**1-1 Submission Process:**

5

**1-4 Interest:**

["Structured debates on controversial topics"]

**1-5 Thoughtful:**

9

**1-6 Supportive:**

8

**1-7 Technical Aspects Versus Position:**

8

**1-8 Gate Keeping:**

8

**1-9 Camera Ready Changes:**

# Reviewer 816x:
As suggested,
W1: We will revise overly optimistic statements to adopt neutral ones, and acknowledge real-world BCI limitations that may also extend to BAC paradigms,
W2, W4, Q3: remove Sec. 5 and expand Sec. 6 with examples from daily living support, rehabilitation, to professional applications,
W3: and further polish the figures with better illustrative content. Thx!

# Reviewer JLRQ:
W1: Due to space constraints, this correspondence from Sec. 5 to Sec. 6 was not made sufficiently explicit. In revision, we will highlight that Components 2 and 3 in Sec. 6.2 are designed to address Challenge 1, while several subsections in Sec. 6.1 corresponds to Challenges 2, 3, and 4. We recognize that current version may seem fragmented, obscuring these connections. Therefore, we will supplement discussions, reorganize the content, and use clear annotations.
Q1: We will add an "Ethics-level" evaluation dimension assessing adherence to ethical principles and user rights protection. In "Empirical Validation Methods," we'll introduce "Ethical Red Teaming" to proactively test vulnerabilities through scenarios challenging ethical safeguards. Thanks.

# Reviewer 4VHB:
W1: (1) In revision, we will strengthen the connection between these alternative views and our arguments, ensuring better coherence. (2) Similarly, for Evaluation Protocol, we will provide better contextualization by emphasizing its necessity first, rather than directly presenting individual model performance metrics.
(3) Regarding ethical consideration, the reviewer's observation is right, as existing frameworks typically incorporate ethical principles as foundational elements already. Our emphasis is on risk prevention, such as safeguarding against adversarial prompts that could lead to information leakage, etc.
(4) Regarding signposting and thesis reinforcement, we will enhance the clarity of our argumentation flow and more explicitly connect each section's evidence to our central views. Thanks.

**3-1 Review Response1:**

Reviewer 816x

**3-2 Reaction To Review1:**

W1: We will revise overly optimistic statements to adopt neutral ones, and acknowledge real-world limitations of existing BCIs that may also inherit to BAC paradigms, such as low information transfer rates, extensive user-specific calibration demands, signal quality concerns, etc.
W2, W4, Q3: Thanks for the suggestion. We will remove Sec. 5 and expand Sec. 6 with examples from daily living support, rehabilitation, to professional applications. Each scenario incorporates core BAC mechanisms, which will be detailed in revision.
W3: We will further polish the figures with better illustrative content.


Q1.
(1) Enhancing high-quality signal acquisition is essential, particularly for non-invasive cases, as this is critical for subsequent analysis. (2) Techniques such as multimodal information fusion should also be strengthened, which may not be well captured in traditional BCIs and unimodal LLMs. Additionally, continuous tracking and real-time processing capabilities need reinforcement to enhance BAC analytical capacity and improve operational stability. These advances will address conventional BCI limitations. (3) Lastly, infrastructure advances such as recent bidirectional adaptive interfaces will help provide significant improvements in communication efficiency and energy consumption. We will further discuss these aspects in detail.

Q2.
Most existing approaches still rely on LOO cross-validation, while LOSO reveals more substantial performance degradation due to within-subject overfitting. This may result from both limited real-world data availability and limitations in current learning methodologies. Additionally, observations indicate significant performance gaps between laboratory and real-world settings, such as P300-based systems achieving 90-98% accuracy in controlled environments V.S. 70-80% in practical applications. We will incorporate more detailed and objective discussion on this matter. We appreciate the reviewer's valuable feedback.

**3-3 Review Response2:**

Reviewer JLRQ

**3-4 Reaction To Review2:**

W1.
We sincerely appreciate Reviewer JLRQ for the thoughtful feedback. Our original intention was for the content in Section 6 to address the challenges outlined in Section 5. However, we acknowledge that due to constraints on length and coverage, this correspondence was not made sufficiently explicit.
In our revision, we will thoroughly address this. Specifically, we will highlight that Components 2 and 3 in Section 6.2 are designed to address Challenge 1, while the "Ethical Safeguards and Data Governance" and "Supervisory Control" subsections in Section 6.1 correspond to Challenges 2, 3, and 4. We recognize that the current presentation may seem fragmented, obscuring these connections. Therefore, we will supplement the discussions, reorganize the content, and use clear annotations in the revised manuscript to make the relationship between Section 5 and Section 6 unequivocal. Thank you for this valuable suggestion.


Q1.
We are grateful to the reviewer for this constructive suggestion. We will incorporate an "Ethics-level" evaluation dimension into our framework.
Specifically, this dimension will assess the system's adherence to established ethical principles and its capacity to protect user rights throughout the collaborative process. It will evaluate key aspects such as data privacy, user autonomy, system transparency, and fairness. Furthermore, within the "Empirical Validation Methods" section, we will introduce the method of "Ethical Red Teaming". This approach proactively tests for vulnerabilities by designing scenarios to challenge the system's ethical safeguards. For example, it can involve attempting to elicit biased responses, tricking the agent into revealing sensitive information, or creating situations where ethical considerations might be broken. We will provide a detailed elaboration of these additions in the revised paper. Thank you.

**3-5 Review Response3:**

Reviewer 4VHB

**3-6 Reaction To Review3:**

W1.
We thank Reviewer 4VHB for constructive feedbacks. (1) The two alternative perspectives we present are intended to critically examine potential risks and harms inherent in BAC technology and its applications. Our responses aim to provide comprehensive, objective countermeasures and mitigation strategies. In the revision, we will strengthen the connection between these alternative views and our primary arguments, ensuring better coherence throughout the discussion. (2) Similarly, for Evaluation Protocol, we will provide better contextualization by emphasizing its necessity first, rather than directly presenting individual model performance metrics.
(3) Regarding ethical consideration part, the reviewer's observation is right, as existing frameworks typically incorporate ethical principles as foundational elements already. Our emphasis is on systematic risk prevention during the initial model design phase, such as safeguarding against adversarial prompts that could lead to information leakage or the generation of harmful content through prompt manipulation.
(4) We appreciate the reviewer's suggestions regarding signposting and thesis reinforcement. In revision, we will enhance the clarity of our argumentation flow and more explicitly connect each section's evidence to our central views.

Q1.
(1) Our primary position is that we could leverage LLM-based agents to enhance traditional BCI systems.
(2) Individuals can contribute through technical innovations, while orgs establish governance frameworks and standards.
(3) We believe privacy should be emphasized more in BAC. As an intelligent assistive technology and the focus is whether it genuinely protects and serves users.
(4) Current work focuses mainly on task completion. Frameworks explicitly balancing privacy protection with complex task execution remain underdeveloped, indicating a valuable research direction.

We will also update the manuscript to address the withdrawn paper citation you mentioned. Thank you.

---

### Meta-Review · Area_Chair_uD56 · 2025-09-12

**Rating:** 7
**Confidence:** 4

**Strengths:**

The authors proposed an appropriate topic for the NeurIPS community to discuss, which is novel and very interesting.

**Weaknesses:**

A few typos, and the structure of the paper needs to be optimized after acceptance.

**Questions:**

See reviewers' questions

**Ethics:**

no issues

**Thoroughness:**

3

---

### Decision · Program_Chairs · 2025-09-26

Accept